# Inequalities and trends in Neonatal Mortality Rate (NMR) in Ethiopia: Evidence from the Ethiopia Demographic and Health Surveys, 2000–2016

Gebretsadik Shibre[1]*, Dina Idriss-Wheeler[2], Sanni Yaya[3,4]

1 Department of Reproductive, Family and Population Health, School of Public Health, Addis Ababa University, Addis Ababa, Ethiopia, 2 Faculty of Health Sciences, Interdisciplinary School of Health Sciences, University of Ottawa, Ottawa, Canada, 3 Faculty of Social Sciences, School of International Development and Global Studies, University of Ottawa, Ottawa, Canada, 4 The George Institute for Global Health, The University of Oxford, Oxford, United Kingdom

* gebretsh@gmail.com

## Abstract

### Background

Substantial inequality in neonatal mortality rates (NMR) remains in low- and middle-income countries to the detriment of disadvantaged subpopulations. In Ethiopia, there is a dearth of evidence on the extent and trends of disparity in NMR. This study assessed the socioeconomic, residence and sex-based inequalities in NMR, as well as examined its change over a sixteen year period in Ethiopia.

### Methods

Using the World Health Organization's (WHO) Health Equity Assessment Toolkit (HEAT) software, data from the Ethiopia Demographic and Health Surveys (EDHS) were analyzed between 2000 and 2016. NMR was disaggregated by four equity stratifiers: education, wealth, residence and sex. In addition, absolute and relative inequality measures, namely Difference, Population Attributable Risk (PAR), Ratio, Relative Concentration Index (RCI) and Slope Index of Inequality (SII) were calculated to understand inequalities from different perspectives. Corresponding 95% Uncertainty Intervals (UIs) were computed to measure statistical significance.

### Findings

Large educational inequalities in NMR were found in 2000, 2005, and 2011, while wealth-driven inequality occurred in 2011. Sex disparity was noted in all the surveys, and urban-rural differentials remained in all the surveys except in 2016. While socioeconomic and area-related inequalities decreased over time, sex related inequality did not change during the period of study.

**Data Availability Statement:** The data underlying the results presented in the study are available from WHO HEAT software webpage: https://www.who.int/gho/health_equity/assessment_toolkit/en/

**Funding:** The author(s) received no specific funding for this work.

**Competing interests:** The authors have declared that no competing interests exist

## Conclusions

NMR appeared to be concentrated among male newborns, neonates born to illiterate and poor women and those living in rural settings. However, the inequality narrowed over time. Interventions appropriate for different subpopulations need to be designed.

## Background

A staggering 2.5 million newborns died, globally, of preventable or treatable causes within the first 28 days of life in 2018 [1]. This represented approximately 63% of all infant deaths and 47% of deaths in children five years and under [1]. Sub-Saharan Africa (41%) and Southern Asia (37%) accounted for close to 80% of the 2018 global estimates for neonatal mortality [1], where interventions to prevent these deaths are minimal [2–4]. Expansion of effective interventions to all who need it can improve survival rates of neonates, however, launching strategies necessary to meet this aim remain a challenge in these regions [5]

Prior research showed socioeconomic inequity in neonatal deaths tended to occur in disadvantaged subpopulations [5]. The majority of deaths took place in Low and Middle Income Countries (LMIC) [6], disfavouring the poor, illiterate and rural communities [5,7]. Interestingly, over the last 20 years, the inequality in NMR has been decreasing in most of the LMICs [5]. As the global community continues to work toward an equity-oriented international agenda, to meet the United Nations' Sustainable Development Goals (SDG), it is imperative that evidence-based interventions reach the disadvantaged subpopulations to alleviate NMR disparities.

In 2018, nearly 100,000 babies died within the first 28 days following birth in Ethiopia [1], contributing to over 50% of the under-five mortality rate in the country. McKinnon et al. (2014) flagged Ethiopia as one of two countries where "wealth related inequality increased by more than 1.5 neonatal deaths per 1000 live births per year between 2000 and 2011" (5, p. 165). This underscores the importance of further research to investigate NMR relative to various dimensions of inequality (wealth, education, residence and sex) and how the disparity has changed over the past nearly two decades in Ethiopia.

Few research endeavors have attempted to assess NMR inequality in Ethiopia [5,7,8]. Our study enriched available evidence in a number of areas. First, it investigated NMR inequality through the rigorous method recommended by the World Health Organization (WHO) Handbook on health inequality monitoring. This method considers different dimensions of inequality measures in the analysis to gain a better understanding of the NMR disparity [9]. Commonly used absolute (difference) and relative (ratio) simple measures of inequality were calculated to reflect the magnitude of the difference in NMR between studied subgroups and how inequality changed over time [9]. In addition to the simple measures, which allow comparison between two subgroups only, the study also estimated the inequality using complex measures that account for all subgroups of an equity stratifier [9]. Second, it used the common equity stratifiers recommended by the WHO [9] which is in contrast to most prior studies that limited NMR inequality analysis to socioeconomic dimensions (i.e. wealth and education) [5,7]. Third, previous studies used traditional regression methods [8,10,11] that do not follow the WHO recommendation for inequality studies [9] or were limited to certain specific areas in the country without including the national context [11,12]. Finally, it assessed NMR inequality for the four Ethiopia Demographic and Health Surveys (EDHS 2000, 2005, 2011, 2016) and was not limited to the first and the last EDHS rounds, unlike prior studies [5,13].

This contributed to understanding the dynamics of NMR differentials in Ethiopia over sixteen years.

The study was designed to answer two questions: 1) what is the status of NMR inequality in Ethiopia across different equity stratifiers?; and 2) how have levels of NMR inequality in different equity stratifiers changed over time between 2000 and 2016?

## Methods

### Study setting

With 109 million people, Ethiopia is the most populous country in Africa second only to Nigeria (14). Ethiopia experienced consistent economic growth over the last decade with agriculture as the major driver [14]. Despite the significant reduction in the proportion of the population living in severe poverty [14], Ethiopia is still one of the poorest nations in the world, with an average per capita Gross Domestic Product (GDP) of approximately US$ 830 in 2017 [15]. According to the Human Development Index (HDI), Ethiopia ranks 174 out of 188 countries, performing poorly in social and economic dimensions such as life expectancy, literacy, education, and a decent standard of living [16].

Ethiopia suffers the double burden of communicable and non-communicable diseases (NCD). In 2017, communicable, maternal, neonatal and nutritional related diseases were the major causes of deaths, while NCDs were the main contributors to disability [17]. Maternal and neonatal mortalities in Ethiopia are one of the highest globally [1,18]. Although improvements in maternal health services are evident [19], thousands of mothers continue to die from pregnancy related problems, driven partly by the inequitable distribution of services favouring advantaged subpopulations [20,21]. The country prepared and began implementing the 2015–2020 Health Sector Transformation Plan (HSTP) aimed at improving maternal and child health through removal of health disparities [22]. Centered on equity of health care, the plan is working towards achieving the Sustainable Development Goals (SDG).

### Data sources

Ethiopia Demographic and Health Surveys conducted in 2000, 2005, 2011 and 2016 were used. The methodology DHS follows has been described elsewhere [23–26]. Briefly, a two-stage cluster design was used to select women aged 15 to 49 years. DHS surveys are nationally representative and collect information and data on a wide range of public health related topics and indicators such as maternal health services, child health, maternal and childhood mortality, socioeconomic status, family planning and domestic violence. They were carried out with the financial and technical assistance of Inner City Fund (ICF) International and provisioned through the USAID-funded MEASURE DHS program.

### Selection of variables

Inequality was measured for the NMR which is defined as the probability (expressed as a rate per 1000 live births) of a neonate dying within the first month of life. In the DHS, birth histories collect information on the date of birth and age of death of the neonate. Data on live births that occurred 5 years preceding the surveys were included in the analysis.

### Measures

NMR inequality was measured over the four time periods using four equity stratifiers: economic status, education, residence and sex. Economic status was approximated by a wealth index computed based on household assets and characteristics of the household. Wealth index

is classified by DHS as poorest, poor, middle, rich and richest and computed using Principal Component Analysis (PCA) [27]. Educational status of the mother was categorized as no education, primary and secondary education. Residence was classified as urban vs. rural and sex as male vs. female.

## Statistical analysis

The inequality in NMR was examined in two steps. First, the NMR was disaggregated by the equity stratifiers mentioned above. Second, the inequality was assessed using five measures of inequality: Difference, Population Attributable risk (PAR), Ratio, Relative Concentration Index (RCI), and Slope Index of Inequality (SII). While Difference and Ratio are simple measures, the other three are complex measures. Whereas Ratio and RCI are relative summary measures, the remaining are absolute summary measures. There is strong evidence suggesting the scientific importance of adopting both absolute and relative summary measures in a single health inequality study [9]. The main reason being, relative and absolute inequality measures may lead to different, even contrasting conclusions [9], and failing to showcase these differing scenarios can potentially bias informed decisions.

Unlike simple measures, complex measures account for the number of categories in a variable. When a population shift is likely to occur, especially when trend analysis is integral to the study, complex measures are likely to reflect the true change in equality over time [9]. On the other hand, simple measures are easy for interpretation and understanding. Therefore, an inequality study combining simple and complex, as well as relative and absolute, measures provides a more comprehensive analysis.

The publicly available WHO's HEAT version 3.1 software was used for the analysis [28]. The procedures followed for calculating summary measures are detailed in the HEAT software technical notes [28], and in the WHO handbook on health inequality monitoring [9]. For education, economic status, and residence, the Difference was calculated as NMR in the poorest group minus and the richest group, no education group minus secondary education group, and rural minus urban populations, respectively. Ratio was calculated as NMR in one category divided by NMR in the other category, using the same groups described for the Difference calculation. No inequality existed if Difference was zero and Ratio was one.

PAR was computed as the difference between NMR estimate for the reference subgroup, *yref*, and the national average of NMR. In this study, yref referred to the following to calculate NMR inequality for PAR: urban setting for a place of residence, female for sex, secondary education for education and richest subgroups for economic status. While zero indicated absence of inequality, the greater absolute value of PAR indicated a higher level of inequality. The SII was computed through a generalized linear regression model. The computation was restricted to two dimensions (education and economic status) and required ranking of a weighted sample of the whole population in order from the most disadvantaged (rank 0) to the most advantaged (rank 1) subgroups. The poorest and uneducated individuals were considered the most disadvantaged, while those that had completed secondary education and the richest subgroups were deemed most advantaged. NMR was regressed against the midpoint value for the wealth or education subgroup and predicted values of the NMR were calculated for those at the two extremes. The difference in the estimated values between rank 0 and rank 1 produced the SII, representing the difference between lowest and highest while considering all subgroups. A zero value for SII indicated an absence of inequality while positive values suggested higher prevalence of NMR in the disadvantaged subgroups. Finally, the RCI measured the extent to which NMR was concentrated among the advantaged or disadvantaged groups. If RCI was zero, there was no inequality in NMR. Negative values indicated concentration of NMR

among the disadvantaged, while positive values showed concentration in advantaged sub-populations.

The change in NMR over time was examined by referring to the 95% UIs of the different survey years. When the UIs did not overlap, there was a statistically significant difference in NMR between any two consecutive years. If the UIs overlapped, then no inequality existed

### Ethical consideration

The study used publicly available data from DHS. Ethical procedures were the responsibility of the institutions that commissioned, funded, or managed the surveys. All DHS surveys were approved by ICF international as well as an Institutional Review Board (IRB) in the respective country to ensure that the protocols were in compliance with the U.S. Department of Health and Human Services regulations for the protection of human subjects.

### Results

Table 1 presents NMR disaggregated by the four dimensions of inequality for each of the four EDHS. In 2000, the highest NMR was recorded among the poor, middle and rich subgroups, whereas the lowest was observed among the poorest and richest. This uneven distribution of NMR resulted in an inverted U shape of wealth-related inequality. In 2016, NMR was dispro-portionately high amongst the rich, with no statistically significant differences between the other four sub-categories. Individuals who completed secondary education experienced the lowest NMR in 2000 and 2005. However, in 2016, NMR was more evenly distributed across all the three subgroups of education. Across all surveyed years, a significant gender related dispar-ity persisted to the disadvantage of male neonates.

Inequality of NMR was analyzed with relative and absolute measures, using simple and com-plex techniques, as presented in Table 2. Wealth (i.e. economic status) related inequality differed widely by the type of inequality measures. The simple measure of Difference indicated no wealth-driven NMR disparity in all the surveys except in 2011, where, on average, an excess of 13 neonatal deaths per 1000 live births were recorded for the poorest subgroups. The SII showed that wealth-based inequality remained in 2011 and 2016. The inequality shifted from the richer subgroups in 2011 (SII = -18; -28, -9) to the poorer ones in 2016 (SII = 11; 2.3, 20). The PAR showed a wealth-related disparity in favour of the wealthier households during the first three sur-veys as the poorest households experienced higher NMR. In 2000, on average, the national NMR would have been reduced by 14 neonatal deaths per 1000 live births (PAR = -14; -15, -12) had all the wealth quintiles been on the par with the richest fifth quintile. Interestingly, the inequality significantly declined over time until it disappeared in 2016.The RCI measure also confirmed NMR inequality in 2011, with the poor enduring most of the neonatal deaths (Table 2).

Both the relative and absolute measures indicated that NMR appeared to be largely concen-trated among infants of women with no formal education. In 2000, the Ratio showed that NMR among the uneducated women was about 2.5 times higher than that for women who had completed secondary education. PAR and RCI also showed clear disparity in NMR, again disfavouring the illiterate subgroup. By contrast, estimates of the SII revealed that NMR was largely prevalent among women with higher educational attainment in both 2000 and 2011. While educational inequality in NMR had been falling over time based on PAR and D mea-sures, other summary measures showed fluctuations in the results (Table 2).

While no urban-rural differentials of NMR were documented using the simple measures of Ratio and Difference, PAR showed that NMR was higher in rural settings in 2000, 2005, and 2011 but disappeared in 2016. Significant sex disparities were evident as NMR was prevalent among male neonates in all the surveyed time periods by all the measures(Table 2).

**Table 1. Neonatal mortality rate (deaths per 1000 live births) in Ethiopia disaggregated by economic status, education, residence and sex across four time periods (2000, 2005, 2011, 2016).**

| Inequality Dimensions | Categories | Survey Years | | | | | | | | | | | |
|---|---|---|---|---|---|---|---|---|---|---|---|---|---|
| | | 2000 | | | 2005 | | | 2011 | | | 2016 | | |
| | | Estimate | LB | UB | Estimate | LB | UB | Estimate | LB | UB | Estimate | LB | UB |
| Economic Status | Poorest | 41 | 33 | 50 | 39 | 32 | 48 | 50 | 42 | 60 | 36 | 27 | 47 |
| | Poor | 66 | 55 | 79 | 39 | 30 | 49 | 48 | 39 | 59 | 34 | 27 | 42 |
| | Middle | 76 | 63 | 92 | 47 | 39 | 58 | 35 | 28 | 45 | 35 | 27 | 45 |
| | Rich | 61 | 50 | 75 | 45 | 36 | 56 | 39 | 31 | 50 | 47 | 37 | 60 |
| | Richest | 45 | 37 | 54 | 30 | 23 | 40 | 37 | 29 | 48 | 40 | 28 | 57 |
| Education | No-education | 62 | 56 | 68 | 41 | 37 | 46 | 46 | 41 | 51 | 39 | 34 | 45 |
| | Primary | 46 | 34 | 62 | 45 | 35 | 57 | 36 | 29 | 43 | 36 | 27 | 46 |
| | Secondary | 25 | 15 | 41 | 21 | 11 | 38 | 22 | 11 | 42 | 32 | 21 | 50 |
| Residence | Rural | 60 | 54 | 65 | 41 | 37 | 46 | 43 | 39 | 48 | 38 | 33 | 43 |
| | Urban | 47 | 36 | 60 | 34 | 23 | 52 | 41 | 31 | 53 | 41 | 25 | 67 |
| Sex | Female | 49 | 43 | 55 | 33 | 29 | 38 | 34 | 29 | 39 | 26 | 21 | 33 |
| | Male | 67 | 60 | 75 | 48 | 42 | 55 | 51 | 45 | 58 | 49 | 42 | 56 |
| National NMR | | 58.2 | | | 40.6 | | | 42.6 | | | 37.9 | | |

Estimate refers to point estimate for NMR; LB = Lower Bound of the estimate; UB = Upper Bound of the estimate

## Discussion

To our knowledge, this is the first attempt to comprehensively evaluate the nature of NMR disparity in Ethiopia over a sixteen year period. The findings showed inequalities in NMR

**Table 2. Inequality in neonatal mortality rate measured by the different inequality measures across the four dimensions of inequality, between 2000 and 2016 EDHS.**

| Inequality Dimensions | | Survey years | | | | | | | | | | | |
|---|---|---|---|---|---|---|---|---|---|---|---|---|---|
| | Categories | 2000 | | | 2005 | | | 2011 | | | 2016 | | |
| | Measures | Estimate | LB | UB | Estimate | LB | UB | Estimate | LB | UB | Estimate | LB | UB |
| Economic Status | D | -4 | -16 | 8.2 | 8.9 | -3 | 21 | 13 | 0.4 | 26 | -4 | -22 | 13 |
| | PAR | -14 | -15 | -12 | -10 | -12 | -9 | -5 | -6.9 | -4 | 0 | -1.6 | 1.6 |
| | R | 0.9 | 0.7 | 1.2 | 1.3 | 0.8 | 1.8 | 1.4 | 0.9 | 1.8 | 0.9 | 0.5 | 1.3 |
| | RCI | 2.1 | -1.9 | 6.1 | -1 | -6.4 | 4.2 | -7 | -12 | -2 | 4.6 | -2.6 | 12 |
| | SII | 7.6 | -2.9 | 18 | -3 | -12 | 6.5 | -18 | -28 | -9 | 11 | 2.3 | 20 |
| Education | D | 37 | 23 | 50 | 20 | 6.2 | 33 | 24 | 8.8 | 39 | 6.8 | -8.8 | 22 |
| | PAR | -33 | -36 | -31 | -20 | -22 | -17 | -21 | -24 | -18 | -6 | -8.2 | -3 |
| | R | 2.5 | 1.2 | 3.7 | 1.9 | 0.7 | 3.1 | 2.1 | 0.7 | 3.5 | 1.2 | 0.6 | 1.8 |
| | RCI | -5 | -7.7 | -2 | -1 | -4.4 | 3 | -6 | -9.6 | -2 | -2 | -7.6 | 3 |
| | SII | -51 | -71 | -31 | -4 | -17 | 9.1 | -27 | -39 | -14 | -9 | -20 | 2.7 |
| Residence | D | 13 | 0 | 26 | 6.7 | -7.9 | 21 | 1.9 | -9.9 | 14 | -3 | -24 | 18 |
| | PAR | -12 | -14 | -10 | -6 | -8.4 | -4 | -2 | -3.2 | -0.1 | 0 | -1.9 | 1.9 |
| | R | 1.3 | 0.9 | 1.6 | 1.2 | 0.7 | 1.7 | 1 | 0.7 | 1.3 | 0.9 | 0.4 | 1.4 |
| Sex | D | 18 | 9.2 | 28 | 15 | 6.9 | 22 | 17 | 8.7 | 25 | 22 | 14 | 31 |
| | PAR | -9 | -10 | -9 | -8 | -8.2 | -7 | -9 | -9.3 | -8 | -12 | -12 | -11 |
| | R | 1.4 | 1.2 | 1.6 | 1.4 | 1.2 | 1.7 | 1.5 | 1.2 | 1.8 | 1.8 | 1.4 | 2.3 |

Estimate refers to point estimate for NMR; LB = 95% lower bound of the estimate; UB = 95% upper bound of the estimate; D = Difference; PAR = Population Attributable Risk; R = Ratio; RCI = Relative Concentration Index; SII = Slope Index of Inequality

favouring socioeconomically advantaged subpopulations (i.e. wealthier and more educated), female neonates and those living in urban settings.

Patterns of inequality varied widely by the type of inequality measure analyzed over the time period studied. Estimates of PAR showed that wealth-related inequalities in NMR fell steadily throughout the study period until it disappeared in 2016. Between 2000 and 2011, on average, NMR decreased by 9 neonatal deaths per 1000 live births. However, based on estimates of SII, the study found wealth-related inequality in 2011 and 2016, but not in 2000 and 2005. The inequality significantly increased in 2011 compared with levels in 2000 and 2005; it subsequently decreased between 2011 and 2016. The RCI similarly demonstrated the pro-rich scenario of neonatal mortality in 2011.

Even though findings differed by type of measures, the study confirmed that wealth-based inequality generally narrowed over time and even disappeared in 2016 for most measures. The slowly improved coverage of maternal and child health care services over time [19], even if the gap in use of these services between the poor and rich persisted or widened [21], could suggest a possible contribution to the noticeable drop in the wealth-based inequality. Moreover, the region's economic growth over the last decade and the significant fall in the proportion of people living in extreme poverty [14] could be other drivers for decreasing the poor-rich NMR gap over time. Our finding that NMR disproportionately impacts the poor was aligned with some prior work [29] but not with others [13].

NMR was concentrated among neonates born to women with no education, coinciding with findings from available literature [5]. Educational inequality in NMR either fell steadily or remained unchanged depending on the type of summary measures used. The 2016 survey year appeared to be the most equitable, where NMR was equally prevalent among the three subgroups of education by most measures of inequality. A notable exception was the resulting SII educational inequality in 2000 and 2011, favouring the uneducated subpopulation. Conflicting with the findings obtained through other measures and from prior studies [5], this may be anomalous, requiring further study.

Large urban-rural disparity in NMR appeared during the first three EDHS time points. However, it had been consistently decreasing throughout the studied years until it disappeared in 2016. Between 2000 and 2011, it was possible to reduce the country's NMR between 2 to 12 deaths per 1000 live births (on average) had the rural setting been on par with the urban areas in terms of neonatal mortality reducing interventions. The disproportionately higher clustering of neonatal mortality reducing interventions (such as access to skilled birth attendants) in urban areas can explain the pro-urban profile of NMR in 2000, 2005 and 2011 [24–26].The disappearance of inequality by 2016 may be due to improved coverage of these interventions in rural settings over time [19].This finding was consistent with prior studies that confirmed the pro-urban nature of NMR [30] but not with others [31].

The relative and absolute sex inequalities were notable. NMR was prevalent among male neonates in all the surveyed periods. In 2016, when sex-based disparity was most pronounced, male neonates experienced, on average, an excess of 22 deaths per 1000 live births and endured about twice as many deaths when compared with female neonates. Unlike other equity stratifiers based inequality, sex-based disparity did not improve with time. Other studies have found similar NMR inequality results to the advantage of female as compared to male neonates [31]. Why, then, are male neonates at higher risk of dying than female neonates? Yet another aspect to explore in future studies.

The nature of NMR disparities in our study can be analyzed using the "inverse equity hypothesis" proposed by Victora CG et al (2000) [32]. The hypothesis provides an explanation for the trends in health inequity, over time, between the poor and rich, and postulates that inequities become worse before getting better. It suggests that during initial scale up of public

health interventions, there is higher uptake and earlier adoption among the rich subgroups, before the poorer subgroups even start utilizing the services. With time, the morbidities and mortalities decline among the rich while the poorer subgroups have not yet fully begun to use the services. Subsequently, the morbidities and mortalities of poorer subgroups are not seen to decrease at the level experienced by the rich. This difference in uptake of interventions between the two groups during the early period leads to the wider inequities in health indicators. Once the interventions reach the poor, the rich have already achieved the "minimum acceptable level" of the unfavourable indicator of interest, like NMR, thus inequities between the two groups narrow in terms of those indicators.

Congruent with the proposed hypothesis, utilization of maternal health care services, such as antenatal care or skilled delivery, has been disproportionately higher for subgroups who are socioeconomically better-off and live in urban areas in Ethiopia [28]. Between 2000 and 2011, these services were used mainly by the rich, more educated and urban populations while the poorer, uneducated individuals living in rural settings did not use them. Consequently, the NMR was higher among the disadvantaged groups during this period. In 2016, with the scale up of maternal health services in Ethiopia, uptake increased among the disadvantaged subgroups leading to decreasing NMR. The wealthier subgroups had reached their minimum acceptable level of NMR at this point; thus, the pace of NMR reduction was less, allowing for poorer subgroups lagging behind to close the NMR inequality gap. Available evidence also suggests that scale-up of complementary interventions such as facility delivery can lead to the decrease of neonatal deaths [33]. Given the demonstrated impact of such interventions, the government of Ethiopia may consider expanding some or all of these interventions in order to meaningfully reduce the burden of neonatal death in the country.

Our study offers policy-relevant information regarding the issue of neonatal mortality from the perspective of the socioeconomic status of the people. Equitable economic policies need to be an integral element of health policies in the country in order to benefit the poor and middle-class sub populations and increase their use of essential maternal health care services. This, in turn, helps narrow the poor-rich gap in health outcome indicators, such as NMR, and ultimately eliminate this income driven disparity. Another important policy implication of our work points to the demonstrated influence of education on health.

Education has to be a fundamental element of public health interventions in the country. It is a powerful tool in preventing health care disparities by interrupting the ongoing impact of poverty [34]. The government of Ethiopia can focus on accelerating coverage of at least secondary education to alleviate disparities in NMR between the subpopulations, which, in turn, can translate to the national NMR reduction. By incorporating equitable income and education policies into health policy, the country has the potential to succeed in attaining not just the SDG for NMR, but in reaching other key indicators as well.

The strength of this study is two-fold. First, inequality in NMR was assessed through different summary measures; adopting a number of inequality measures in the study helped showcase various possible patterns of NMR inequality. Second, the findings used an established high quality WHO monitor database prepared by experts, contributing to the quality of evidence analyzed and reported in this paper. However, the study also had some limitations. It did not identify factors that contribute to the measured NMR inequality. Future studies should consider using a decomposition approach to study factors that contribute to the NMR inequality to see whether their contribution changes with time. Furthermore, the EDHS dataset-based findings cannot be applied to areas smaller than the subnational regions and city administrations. Inequality in NMR should be considered in smaller areas such as villages, towns, districts and zones.

## Conclusions

The study found inequalities in NMR across the different equity stratifiers. Encouragingly, however, NMR inequality in Ethiopia has been on the decline according to sixteen years of studied data. The improvement in inequality of NMR over time may be associated with many factors such as Ethiopia's dedication to meet the NMR SDG of 12 deaths per 1000 newborns by 2030, improved economic growth, and improved coverage of maternal and child health services. Inequality studies such as this are useful to inform equity-oriented interventions aimed at eliminating inequality; they signal affected subgroups to strategically target programming and alleviate the problem. Policy makers should prioritize the subpopulations experiencing the highest NMR without ignoring the whole population. Despite gains made so far, much work is still needed to minimize, if possible, eliminate, NMR disparity across different equity stratifiers in Ethiopia.

## Acknowledgments

The authors acknowledge WHO for making this equity analysis software freely accessible.

## Author Contributions

**Conceptualization:** Gebretsadik Shibre.

**Formal analysis:** Gebretsadik Shibre.

**Methodology:** Gebretsadik Shibre.

**Writing – original draft:** Gebretsadik Shibre.

**Writing – review & editing:** Gebretsadik Shibre, Dina Idriss-Wheeler, Sanni Yaya.

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
