## [Decision Letter · Decision Letter 0]

6 Apr 2020

PONE-D-19-35043

Inequalities and Trends in Neonatal Mortality Rate (NMR) in Ethiopia: Evidence from the Ethiopia Demographic and Health Surveys, 2000-2016

PLOS ONE

Dear mr Shibre,

Thank you for submitting your manuscript corresponding dpt to PLOS ONE. After careful consideration, we feel that it has merit but does not fully meet PLOS ONE’s publication criteria as it currently stands. Therefore, we invite you to submit a revised version of the manuscript that addresses the points raised during the review process.

The manuscript reads well. However, it lacks some methodological weakness therefore needs further analyses, literature review and  appropriate discussion. 

We would appreciate receiving your revised manuscript by May 21 2020 11:59PM. To enhance the reproducibility of your results, we recommend that if applicable you deposit your laboratory protocols in protocols.io, where a protocol can be assigned its own identifier (DOI) such that it can be cited independently in the future. For instructions see: http://journals.plos.org/plosone/s/submission-guidelines#loc-laboratory-protocols

We look forward to receiving your revised manuscript.

Kind regards,

Mahfuzar Rahman, MD, PhD

Academic Editor

PLOS ONE

Journal Requirements:

The name of the colleague or the details of the professional service that edited your manuscriptA copy of your manuscript showing your changes by either highlighting them or using track changes (uploaded as a *supporting information* file)

3. Our internal editors have looked over your manuscript and determined that it is within the scope of our Health Inequities and Disparities Research Call for Papers. This collection of papers is headed by a team of Guest Editors for PLOS ONE: Clare Bambra, Hans Bosma, Diana Burgess, Joseph Telfair, Barbara Turner, and Jennie Popay. The Collection will encompass a diverse range of research articles on health inequities and disparities.  Additional information can be found on our announcement page: hhttps://collections.plos.org/s/health-inequities

If you would like your manuscript to be considered for this collection, please let us know in your cover letter and we will ensure that your paper is treated as if you were responding to this call. If you would prefer to remove your manuscript from collection consideration, please specify this in the cover letter.

Reviewers' comments:

Reviewer's Responses to Questions

**Comments to the Author**

1. Is the manuscript technically sound, and do the data support the conclusions?

Reviewer #1: Yes

Reviewer #2: Yes

2. Has the statistical analysis been performed appropriately and rigorously? 

Reviewer #1: Yes

Reviewer #2: Yes

3. Have the authors made all data underlying the findings in their manuscript fully available?

Reviewer #1: Yes

Reviewer #2: Yes

4. Is the manuscript presented in an intelligible fashion and written in standard English?

Reviewer #1: Yes

Reviewer #2: Yes

5. Review Comments to the Author

Reviewer #1: The manuscript is well written and well presented. The analysis is perfect. It can be published with or without my suggestions. I have made one simple suggestion to add the overall NMR in one row in table-1.

Reviewer #2: The manuscript sheds light on an important topic. The current format of the manuscript was not suitable for review as there is not page or line numbers. My following major comments refer to page/paragraph numbers based on the submission document:

Introduction:

1. p.11, lines 13‒15: Authors need to elaborate the statement “(previous) regression methods not suitable for inequality studies”, especially when HEAT itself uses regression models for estimating RII and SII.

Results:

2. p.22, Table 2: The table is hard to read and needs proper notes for abbreviated forms if inequality measures.

3. Authors may consider using charts, which would be useful to compare the confidence intervals for categories over time.

Discussion:

4. p.23‒24: Does your findings support the first para of discussion section “The findings showed evident inequalities in NMR favoring socioeconomically advantaged subpopulations (i.e. wealthier and have higher education), female children and those living in urban settings”? If you compare the confidence intervals, NMR disparity has remained similar over time for all the dimensions except education.

5. The findings of this study clearly indicates towards the ‘inverse equity hypothesis,’which states that as countries scale up use of health services from initially low levels, early increases in inequality could be expected—if the hypothesis holds true, after the initial widening, inequalities in these countries should begin to narrow. The authors may want to review papers by Victora et al. (2000) [doi: 10.1016/S0140-6736(00)02741-0] and Cutler, Deaton & Lleras-Muney (2006) on this issue.

6. The authors appear to miss some important articles while comparing their findings with existing studies. I’d recommend to review papers by Onarheim et al. (2012) [doi:10.1371/journal.pone.0041521]; Skaftun, Ali & Norheim (2014) [doi:10.1371/journal.pone.0106460]; and Ambel et al. (2017) [doi: 10.1186/s12939-017-0648-1]; among others.

7. p.24, para 3: “Even though findings differed by type of measures, the study confirmed that wealth driven inequality had generally narrowed….. suggest a possible contribution to the noticeable drop in the wealth-based inequality.” This statement needs to be supported by evidence. Given the starting period(s), country-wide initiatives towards UN SDG or HSTP are unlikely to affect study’s findings.

8. An aim of the HEAT software is to compare inequity over time/countries and disseminate guidance for the policymakers and public health practitioners to address the disparities. I would strong recommend to include a discussion on policy and programmatic implications of the study findings. for this reason, the authors may want to refer to the following published papers: Marchant et al. (2019) [doi: 10.1503/cmaj.190219]; Knippenberg et al. (2005) [doi: 10.1016/S0140-6736(05)71145-4]; Steven et al. (2019) [doi: 10.4314/ejhs.v29 i6.3]; Onarheim et al. (2012) [doi:10.1371/journal.pone.0041521].

6. PLOS authors have the option to publish the peer review history of their article (what does this mean?). If published, this will include your full peer review and any attached files.

Reviewer #1: Yes: Dipak K. Mitra

Reviewer #2: Yes: Karar Z. Ahsan

---

## [Author Response · Author response to Decision Letter 0]

18 Apr 2020

Dear Editor, 

This letter is in reference to your email with reviewers’ comments. We are very pleased that the manuscript is potentially acceptable for publication in PLOS ONE once we have carried out some revisions. 

We would like to thank the reviewers for their insightful and helpful comments and for giving us the chance to revise our manuscript. We believe the revised manuscript has been significantly improved and the reviewers’ comments have been addressed adequately. We think in its current form it will make a valuable contribution to the literature on this increasingly important topic. 

Please find for your kind consideration the followings: 1) a section-by-section response to the comments and suggestions of the reviewers (below) and 2) the revised manuscript provided as a marked-up copy and a clean copy. 

We hope that these changes meet with your favourable consideration. Please do not hesitate to get in touch if you require any further information.

Response to reviewers

Reviewer #1: The manuscript is well written and well presented. The analysis is perfect. It can be published with or without my suggestions. I have made one simple suggestion to add the overall NMR in one row in table-1.

Response: We would like to thank the reviewer for the thoughtful comments and for taking the time to read our manuscript.We agree with the suggestion made which has been applied.

Reviewer #2: The manuscript sheds light on an important topic. The current format of the manuscript was not suitable for review as there is not page or line numbers. My following major comments refer to page/paragraph numbers based on the submission document:

Response: Thanks for pointing out this mistake. This has now been revised.

Introduction:

1. p.11, lines 13‒15: Authors need to elaborate the statement “(previous) regression methods not suitable for inequality studies”, especially when HEAT itself uses regression models for estimating RII and SII.

Response: The reviewer’s point is valid. We have now revised the statement.

Results:

2. p.22, Table 2: The table is hard to read and needs proper notes for abbreviated forms if inequality measures.

Response: The suggestion of the reviewer was considered and applied.

3. Authors may consider using charts, which would be useful to compare the confidence intervals for categories over time.

Response: We thank the reviewer for this relevant suggestion. While drafting the results, we tried to present main findings through charts and/or line graphs. But a major issue has prevented us from doing so. In fact, in the line graphs, the confidence intervals for the different categories (dimensions of inequality) when plotted on same x-y plane are too close to each other and nearly override one another so that readers cannot draw any useful information (see figure below). If we opt to present confidence intervals for the different categories separately, then we would have several charts/graphs. This is not effective and is not in line with the journal requirements. In the new figure below, we presented here a line graph that shows the wealth and education related disparities using SII measure. In 2005 and 2011, intervals for wealth and education are passing over the other and make it difficult to separate out the individual intervals. The problem would even be more complex if we had added intervals for the other categories (residence, sex). Thus, we strongly believe that a table is a better way to highlight the disparities across the various dimensions. 

Fig. wealth and education-based NMR disparity 

Discussion:

4. p.23‒24: Does your findings support the first para of discussion section “The findings showed evident inequalities in NMR favoring socioeconomically advantaged subpopulations (i.e. wealthier and have higher education), female children and those living in urban settings”? If you compare the confidence intervals, NMR disparity has remained similar over time for all the dimensions except education.

Response: The suggestion of the reviewer was considered. In this first paragraph, we only tried to describe whether NMR inequality remains in all dimensions of inequality examined in the study. We did not attempt to discuss whether inequalities changed over time as this is clearly discussed in other paragraphs of the discussion section. The question whether or not confidence intervals overlap is important when exploring changes over time, rather than examing whether disparity existed in the first place. For this latter case, showing that the intervals do not contain the null value is sufficient. For PAR, D, RCI and SII measures, the null value is 0. So, any intervals computed from these measures that crossed 0 shows no inequality. For the R, the null value is 1 and any interval obtained from this measure that crossed 1 do show absence of inequality. This is how we interpreted our findings and is in line with what was described in our methods section. 

To show that your concern was addressed in other paragraphs of the discussion, we discussed inequality for each dimension separately below to show whether these disparities increased or decreased over time and whether intervals overlap or not:

Wealth-based disparity: wealth related disparity appeared by Difference and RCI in 2011 only and we could not talk problem of overlap. By SII, it appeared in 2011 and 2016. Since inequalities are negative in 2011 (-28, -9) and positive in 2016 (2.3, 20), these two confidence intervals do not overlap at all. There is no way for negative and positive intervals to overlap. The PAR showed inequality in 2000, 2005 and 2011 with confidence intervals respectively (-15, -12), (-12, -9) and (-6.9, -4). As we can see, they do not overlap. The only exception is that -12 is both the end of the first interval and is as the same time the starting point for second interval. But, this does not constitute overlap of the intervals. If for instance the second interval is (-14, -11), then we can subjectively declare that the two intervals overlap as they have substantial number of points in common. 

Residence-based disparity: we showed residence related inequality based on the PAR measure for the years 2000, 2005 and 2011. The intervals are (-14, -10), (-8.4, -4) and (-3.2, -0.1) respectively. As we can see, these three intervals are not overlapping. 

Sex-based disparity: we showed sex disparity of NMR by all the three measures. Based on PAR for instance, the four intervals for the year 2000-2016 respectively are -10, -9; -8.2, -7; -9.3,-8 and -12, -11.Intervals in 2000 and 2011, and 2005 and 2011 have only slight overlap. But, intervals in 2000 and 2005& in 2000 and 2016 do not overlap. So as we move from 2000 to 2016, it increased overall.

5. The findings of this study clearly indicates towards the ‘inverse equity hypothesis,’which states that as countries scale up use of health services from initially low levels, early increases in inequality could be expected—if the hypothesis holds true, after the initial widening, inequalities in these countries should begin to narrow. The authors may want to review papers by Victora et al. (2000) [doi: 10.1016/S0140-6736(00)02741-0] and Cutler, Deaton &Lleras-Muney (2006) on this issue.

Response: We agree with this suggestion and the manuscript was revised accordingly. 

6. The authors appear to miss some important articles while comparing their findings with existing studies. I’d recommend to review papers by Onarheim et al. (2012) [doi:10.1371/journal.pone.0041521]; Skaftun, Ali &Norheim (2014) [doi:10.1371/journal.pone.0106460]; and Ambel et al. (2017) [doi: 10.1186/s12939-017-0648-1]; among others.

Response:The suggestion of the reviewer was considered and applied.

7. p.24, para 3: “Even though findings differed by type of measures, the study confirmed that wealth driven inequality had generally narrowed…. suggest a possible contribution to the noticeable drop in the wealth-based inequality.” This statement needs to be supported by evidence. Given the starting period(s), country-wide initiatives towards UN SDG or HSTP are unlikely to affect study’s findings.

Response: The suggestion of the reviewer was considered and applied.

8. An aim of the HEAT software is to compare inequity over time/countries and disseminate guidance for the policymakers and public health practitioners to address the disparities. I would strongly recommend to include a discussion on policy and programmatic implications of the study findings. for this reason, the authors may want to refer to the following published papers: Marchant et al. (2019) [doi: 10.1503/cmaj.190219]; Knippenberg et al. (2005) [doi: 10.1016/S0140-6736(05)71145-4]; Steven et al. (2019) [doi: 10.4314/ejhs.v29 i6.3]; Onarheim et al. (2012) [doi:10.1371/journal.pone.0041521].

Response :Thanks for this excellent suggestion which has now been applied.

---

## [Decision Letter · Decision Letter 1]

28 May 2020

Inequalities and Trends in Neonatal Mortality Rate (NMR) in Ethiopia: Evidence from the Ethiopia Demographic and Health Surveys, 2000-2016

PONE-D-19-35043R1

Dear Dr. Shibre,

We are pleased to inform you that your manuscript has been judged scientifically suitable for publication and will be formally accepted for publication once it complies with all outstanding technical requirements.

With kind regards,

Mahfuzar Rahman, MD, PhD

Academic Editor

PLOS ONE

Additional Editor Comments (optional):

Reviewers' comments:

Reviewer's Responses to Questions

**Comments to the Author**

1. If the authors have adequately addressed your comments raised in a previous round of review and you feel that this manuscript is now acceptable for publication, you may indicate that here to bypass the “Comments to the Author” section, enter your conflict of interest statement in the “Confidential to Editor” section, and submit your "Accept" recommendation.

Reviewer #2: All comments have been addressed

2. Is the manuscript technically sound, and do the data support the conclusions?

Reviewer #2: (No Response)

3. Has the statistical analysis been performed appropriately and rigorously? 

Reviewer #2: (No Response)

4. Have the authors made all data underlying the findings in their manuscript fully available?

Reviewer #2: (No Response)

5. Is the manuscript presented in an intelligible fashion and written in standard English?

Reviewer #2: (No Response)

6. Review Comments to the Author

Reviewer #2: (No Response)

7. PLOS authors have the option to publish the peer review history of their article (what does this mean?). If published, this will include your full peer review and any attached files.

Reviewer #2: Yes: Karar Z. Ahsan

---

## [Editor Report · Acceptance letter]

1 Jun 2020

PONE-D-19-35043R1 

Inequalities and Trends in Neonatal Mortality Rate (NMR) in Ethiopia: Evidence from the Ethiopia Demographic and Health Surveys, 2000-2016 

Dear Dr. Shibre:

I am pleased to inform you that your manuscript has been deemed suitable for publication in PLOS ONE. Congratulations! Your manuscript is now with our production department. 

With kind regards,

on behalf of

Dr. Mahfuzar Rahman 

Academic Editor

PLOS ONE